# Optimization of Manufacturing Parameters and Tensile Specimen Geometry for Fused Deposition Modeling (FDM) 3D-Printed PETG

**DOI:** 10.3390/ma14102556

**Published:** 2021-05-14

**Authors:** Arda Özen, Dietmar Auhl, Christina Völlmecke, Josef Kiendl, Bilen Emek Abali

**Affiliations:** 1Chair of Polymer Materials Science and Technologies, Institute of Material Science and Technology, Technische Universität Berlin, Ernst-Reuter-Platz 1, 10587 Berlin, Germany; auhl@tu-berlin.de; 2Stability and Failure of Functionally Optimized Structures Group, Institute of Mechanics, Technische Universität Berlin, Einsteinufer 5, 10587 Berlin, Germany; christina.voellmecke@tu-berlin.de; 3Department of Civil Engineering and Environmental Sciences, Institute of Engineering Mechanics and Structural Analysis, Universität der Bundeswehr München, Werner-Heisenberg-Weg 39, 85577 Neubiberg, Germany; josef.kiendl@unibw.de; 4Division of Applied Mechanics, Department of Materials Science and Engineering, Uppsala University, Box 534, SE-75121 Uppsala, Sweden; bilenemek@abali.org

**Keywords:** additive manufacturing, 3D printing, mechanics, slicing approach, polymers, finite element method

## Abstract

Additive manufacturing provides high design flexibility, but its use is restricted by limited mechanical properties compared to conventional production methods. As technology is still emerging, several approaches exist in the literature for quantifying and improving mechanical properties. In this study, we investigate characterizing materials’ response of additive manufactured structures, specifically by fused deposition modeling (FDM). A comparative analysis is achieved for four different tensile test specimens for polymers based on ASTM D3039 and ISO 527-2 standards. Comparison of specimen geometries is studied with the aid of computations based on the Finite Element Method (FEM). Uniaxial tensile tests are carried out, after a careful examination of different slicing approaches for 3D printing. We emphasize the effects of the chosen slicer parameters on the position of failures in the specimens and propose a simple formalism for measuring effective mechanical properties of 3D-printed structures.

## 1. Introduction

Additive manufacturing is a production process relying on building structures layer-by-layer by utilizing the following strategy. First, the computer-aided design (CAD) of a structure is created by a design software. Second, this CAD model is converted by software, called a slicer, to a 3D printing code, where process information is supplied for manufacturing [1,2]. This strategy is taken for granted and used in different branches of the industry, such as automotive, aeronautics, and biomechanical [3,4], as well as in research, especially for studying metamaterials [5,6,7]. Additive manufacturing processes, according to ISO/ASTM52900-15, are categorized in one of the following seven groups: material extrusion, material jetting, binder jetting, sheet lamination, vat photopolymerization, powder bed fusion, and directed energy deposition [8,9,10].

Fused deposition modeling (FDM) is an additive manufacturing method based on materials extrusion [1]. For thermoplastic polymers in the form of spools of filaments, one filament is pushed through a nozzle at slightly over the melting temperature. In this way, the layer is deposited as a viscous fluid, solidifying by decreasing its temperature under convective heat transfer on the surface. This procedure is repeated layer-by-layer, allowing almost any shape. Properties and the final quality depends on material chemistry [11], as well as chosen process parameters in manufacturing [12,13]. Affected by the layer-by-layer production, an inner structure occurs in the final product, leading to structure-related anisotropic properties [14,15], as well as higher-order (so-called size) effects [16,17,18]. Estimating mechanical behavior of materials is of interest especially regarding production parameters [19,20] and used algorithms for depositing layers. For the mechanical characterization of parts manufactured by the FDM, no standard methods are established [21,22].

Ample studies have been done for modeling polymers manufactured by the FDM. Among others, structurally dependent elasto-plastic behavior has been modeled [23]. Theoretical models are proposed for determining tensile strength and Young’s modulus with different raster angles and layer thicknesses [24]. Different length scales and their interactions have been investigated—mainly based on the classical laminate theory (CLT) [25]. However, the classical laminate theory uses assumptions with limited validity for FDM, such as perfect bonding [26]. Some FDM materials are compared by performing tensile experiments [27]. Even differences between characterization methods for polymers (tensile testing) are investigated [28]. Effects of different tensile test specimen geometries taken from ASTM D638 (Standard Test Method for Tensile Properties of Plastics) on the anisotropy have been studied regarding processing parameters, such as the raster pattern, print orientation, and tensile specimen dimensions [29]. Layer thickness and build orientation are analyzed by tensile, flexural, and impact tests [30]. Polymer composites for FDM are proposed and characterized with polypropylene (PP) and natural fibers, such as hemp (Cannabis sativa) or harakeke (Phormium tenax) [31]. Different layer orientations of acrylonitrile butadiene styrene (ABS) polymer [32] demonstrate increased tensile strength along fibers in the FDM [33]. It is not only layer thickness, but also orientation angle and air gap that affects the mechanical properties of polymers greatly [34]; this phenomenon has been identified [35], and for details we refer to [19]. A correlation between 3D-printing time and dimensional accuracy is established by parameter optimization [36]. For achieving high visual quality and fast 3D printing, the slicing algorithm is optimized [37]. The effect of printing time on mechanical properties of FDM 3D-printed Poly(Lactic Acid) (PLA) [38] and PLA/Graphene composites are investigated [39].

Determination of material properties by using a uniaxial tensile test is challenging in 3D-printed materials. ASTM D638 is designed for plastics, but the suggested design (topology) causes a premature fail [14]. Moreover, a standard feature called infill patterns manipulates the materials’ response, owing to an inner substructure [40,41,42]. This substructure-related response deviation is examined in Polylactic Acid (PLA) parts with five different infill patterns [43]. Additionally, other process parameters, such as raster layup, including raster angle and width, as well as contour width are investigated [44,45,46] for their effects on the toughness and strength leading to interlocking mechanisms [47]. Build orientation, layer thickness, and feed rate are discussed on 3D-printed PLA samples in [48], and layer thickness and raster angle parameters for PLA and ABS in [49]. In order to model the mechanical response of additive manufactured polymers [50,51,52], well-known homogenization techniques are used in composite materials [53], for example by using a variation of carbon-fiber content in thermoplastic matrix-based composites built by the FDM [54] and also for identifying substructure-related anisotropic properties [55] to be used in computations [56,57].

As we aim for developing a consistent approach for characterizing ultimate tensile strength under uniaxial loading, we emphasize the importance of the selected specimen topology, as well as the process parameters in slicer settings. We study and demonstrate how to obtain adequate repeatability, and thus consistent material parameters. In order to determine the materials’ response, ASTM D638 and ISO 527-2 (determination of tensile properties for molding and extrusion plastics) have been utilized, where some problems were reported [14,31,45] as mainly being effected by prescribed curvatures in the specimen structures and being very challenging to manufacture in 3D printers. As a remedy, ASTM D3039 (Standard Test Method for Tensile Properties of Polymer Matrix Composite Materials) was suggested [14,25,58]. We stress that these issues are partly because of process parameters, and examine different tensile test specimen geometries experimentally, as well as numerically, by Finite Element Method (FEM)-based simulations. Four different specimen configurations have been prepared, namely, two from ASTM D3039 and two based on ISO 527-2. Reliable results with a low standard deviation are obtained by “fine-tuning” the process parameters and modifying the specimen structure, nevertheless still using the suggested ISO standard.

## 2. Materials and Methods

Four different tensile specimen geometries were investigated: ASTM D3039, ASTM D3039 angle, ISO 527-2, and ISO-modified (based on ISO 527-2). The specimen specifications are compiled in Table 1 and their drawings are depicted in Figure 1.

### 2.1. Fused Deposition Modeling

The samples were produced by an FDM-type 3D printer, namely Ultimaker 3 Extended (Ultimaker B.V., Geldermalsen, The Netherlands). White-PETG filaments were purchased from Materials4Print GmbH & Co. KG (Bad Oeynhausen, Germany). The tensile specimens’ CAD models were achieved in open-source platform Salome 9.3 and exported as *stl* files leading to G-codes prepared by Ultimaker Cura 4.3.0 (Ultimaker B.V., Geldermalsen, The Netherlands) with the aid of selecting process parameters, such as slicing speed, layer thickness, temperature, and so forth. These parameters are of utmost importance, and we provide them in Table 2.

All specimens were printed layer-by-layer by choosing the maximum possible filling. All contours are avoided in 3D printing in order to eliminate any destruction of the unidirectional structure. All layers were unidirectional with 0° orientation. For printing the structures, we used line patterns. We emphasize that the fiber is used in the jargon of FDM denoting the 3D-printed line pattern. The line pattern steers the layers and thus generates fibers. These fibers were not connected to the endpoints—the connecting infill lines setting was off, because it could destruct the unidirectionality. The material properties of PETG used for printing are given in Table 3. We assume that these parameters (supplied by the manufacturer) are determined by using an injection mold specimen such that the porosity is expected to be nearly zero (ideal case). In FDM, between filaments, depending on the parameters, voids occur, leading to a porous structure. Hence, the values supplied by the manufacturer are understood as an upper threshold of “effective” parameters obtained from the specimens printed by FDM-based manufacturing.

#### 2.1.1. Slicing Approach

By selecting default settings in the slicer, we observed problems in producing a unidirectional (UD) structure. Hence, we propose two particular changes: development of new travel paths and optimization of slicing sequences.

#### Optimization of Excess Travel Paths

Travel settings are one of the key process parameters in slicer settings. We demonstrate how to adjust these settings for the uniaxial tensile specimen of a UD structure in order to prevent any excess travel lines in specimens. Excess travel lines may cause premature failures, leading to stress concentrations as observed in experiments. In order to make the role of this parameter obvious, we start off with the default travel configuration in Cura (slicer). In default, the slicer tries to minimize the print time such that the continuous production may end up with additional (excess) layers deposited while the nozzle travels. These travel lines are pathways for the nozzle so as to reach a specific position. The default configuration in Cura is generating travel lines inadequate for the UD structure aimed for uniaxial testing, and the suggested layer deposition by the default configuration is illustrated in Figure 2a, making it obvious that the UD structure is difficult to maintain.

Mostly, a standard configuration is obtained by an algorithm optimizing speed or weight. Resulting inner travel lines disturb the aimed for UD structure. Several specimens are manufactured simultaneously in one production run; therefore, inner travel lines are different in each specimen, making a comparative analysis unreliable. Moreover, as seen in Figure 2a, the black lines indicate that the nozzle introduces weak spots within the specimen, and along these inner travel lines, we expect a stress localization and a premature failure of the structure. Obviously, such a failure is not representative of the material itself. An optimized set of settings, specifically for the UD structure, is compiled in Table 4 for establishing a new travel path approach.

#### Optimization of Slicing Sequence

By considering the production time as well, we have succeeded in defining a new configuration setup with parameters given in Table 4 that substantially increases the inner structure by placing travel lines to the outer contour, as demonstrated in Figure 2b. Note that all layers should be produced by the top layer configuration except the bottom layer. In this context, the slicing sequence means the order of the printing areas. The specimens were printed with two different slicing approaches, called Slicing A and Slicing B. In one process, many specimens were manufactured. Their positioning, called layout, is of importance for the Ultimaker Cura slicing algorithm. Figure 3 shows the Slicing A and Slicing B results figuratively.

In Figure 3, the yellow areas present the already deposited material, whereas the gray places show yet to-be-deposited sections. In Slicing B (Figure 3b), the 3D printer deposits in the order given by 1, then 2, and 3. When Area 4 is being manufactured, Areas 1 and 2 nearly attain the room temperature such that the bond in the contact lines of the areas between 1 and 4, as well as 2 and 4, may differ from the rest. Observed premature failures in experiments in these possibly weak contact areas are justified by this deficiency in thermal fusion. Hence, we understand that slicer settings may affect the onset of failure in 3D-printed parts as a consequence of heterogeneous properties caused by the printing strategy. Therefore, we have developed Slicing A as follows.

In Slicing A (see Figure 3a), the area denoted by 1 is initially produced. It is important that the top-right and left-bottom edges are sliced without pausing in Area 1. When Area 1 has been completed, there are two open edges (Areas 2 and 3) in opposite locations. Still, we expect weaker bonds in contact lines between Areas 1 and 2 as well as 1 and 3. However, these weak regions are divided on both ends, contrary to Slicing B, where weak regions are located at the same end. Therefore, we expect that Slicing A will perform better by using the layout given in Table 5.

A new travel path can be used in every geometric shape and even in complex shapes. However, the slicing sequence depends on the geometry. Each geometric shape is to be optimized separately, and default parameters were obtained by an optimization procedure to minimize the printing time by ignoring process-related anisotropy and its possible consequences.

### 2.2. Unidirectional Tensile Tests

Prior to mechanical tests, all specimens were preserved at a 40 °C vacuum oven against the water uptake. Uniaxial tensile tests were performed with a Zwick 1446 (Zwick, Ulm, Germany). testing machine. A mechanical extensometer was utilized to measure strain. We provide the tensile equipment and the test set-up in Figure 4. Experiments were conducted and steered by displacement with a ramp speed of 2 mm/min on ISO527-2, ASTM D3039, and ASTM D3039 angle specimens; and with a ramp speed of 2.5 mm/min on ISO-modified specimens. In order to study the strain rate sensitivity (viscoelasticity), additional experiments were conducted for a sizeable range of strain rates from 0.01 to 100 s^−1^. Postprocessing was performed by the corresponding software leading to values of the ultimate tensile strength (UTS) and Young’s modulus. We provide the number of 3D-printed and tested specimens in Table 6 for assessing the reliability. In total, 48 specimens were 3D-printed and tested.

In order to remove slack from the specimen and test equipment, preloading is recommended prior to the tensile test. Therefore, we applied up to 0.1 MPa at a low rate (quasistatic condition). Strain was corrected by initially setting it to zero. However, we did not correct the stress results, as the modulus calculation involved stress differences. We determined that the material was elastic without any significant rate-dependency at room temperature. We circumvented from determining Poisson’s ratio and used the manufacturer’s value.

### 2.3. Computation by Using the Finite Element Method

Accompanying the experiments, simulations have been utilized for understanding the structural effect on the test results. A standard numerical implementation was used based on [59]. The finite element method (FEM) was employed by solving the so-called weak form:(1)$$\int_{\Omega }\sigma_{ij}\delta u_{j,i}\;dV=\int_{\partial \Omega_{N}}{\hat{t}}_{i}\delta u_{i}\;dA\;,$$
on a computational domain, Ω, with its closure, ∂Ω, as the boundary, which is the image of the underlying continuum body. We understand the Einstein summation convention over repeated indices, where all Latin indices i,j… run from 1 to 3 (*x* to *z*) in Cartesian coordinates. The test function, $\delta u_{i}$, is chosen from the same Hilbertian Sobolev space as the displacement field, $u_{i}$, known as the isoparametric Galerkin procedure [60,61,62,63]. As the deformation on a tensile test is small, we simplify the system and use the linear strain measure,
(2)$$\varepsilon_{ij}=\frac{1}{2}\left(u_{i,j}+u_{j,i}\right)\;.$$

Observed elasticity without rate effects justifies the use of Hooke’s law,
(3)$$\sigma_{ij}=C_{ijkl}\varepsilon_{kl}\;,$$
where the stress tensor, $\sigma_{ij}$, is linearly related to the strain tensor, $\varepsilon_{ij}$, by the stiffness tensor of rank four, $C_{ijkl}$. As we want to distinguish topology (specimen geometry) effects from the slicer caused anisotropy, we model the material as isotropic and homogeneous (ideal bond between fibers and layers) by using the same engineering constants, Young’s modulus, E=2150MPa, and Poisson’s ratio, ν=0.35. For the space discretization, we implement linear form functions and tetrahedron elements with a suitable mesh acquired by an *h*-convergence analysis. All preprocessing steps, that is, CAD generation, boundary conditions marking, and triangulation, have been accomplished in Salome 9.3. Computation is achieved with the aid of open-source codes developed under the FEniCS project [64,65]. This implementation has been verified by closed-form solutions in [66,67].

Standard uniaxial testing simulation was performed for different geometries. On one end, $x_{1}=0$, the specimen is clamped along the axis, $u_{1}=0$. This boundary is of Dirichlet type. On the other end, the specimen is pulled by a given force per area, traction ${\hat{t}}_{i}$, which is a Neumann-type boundary condition applied on $\Omega_{N}$. An illustration is depicted in Figure 5.

Since linear elements for displacement were used, stresses are constant within the elements. Postprocessing was done in ParaView 5.6 where the stresses were smoothed due to use of the same mesh for the sake of better visualization. The accuracy of the computation was obtained by an *a posteriori* error analysis based on the aforementioned convergence analysis. Affected by these computations, preparation of the experiments was made possible, as discussed in the following.

## 3. Results and Discussion

For a comparative study of structure-related effects on stress distribution during a uniaxial tensile test along the *x* axis, we performed simulations with aforementioned geometries, namely, ISO527-2, modified-ISO, ASTM D3039, and ASTM D3039 angles. Normal stress, $\sigma_{xx}$, is used for a qualitative analysis. The stress distribution is expected to be constant within the part of the specimen used for assessment. Results are depicted in Figure 6.

All geometries achieved, as expected, a constant stress within the gauge, away from the clamped ends (tabs). Curvature in the ISO geometry allows for smoother transition by preventing a stress concentration near the edges. Therefore, we designed ISO-modified geometries with greater curvatures. This minor modification helps to manufacture a specimen, where a failure within the transition zone is prevented. After a complete tensile experiment until failure, a macroscopic crack is initiated in the appropriate region (gauge region) as seen in Figure 7a. In ASTM D3039 geometries, stress localization between the tab and gauge sections causes a premature failure, as demonstrated in Figure 7b.

An analogous procedure then follows by adding a curvature to ASTM D3039 specimens. ASTM D3039 angle specimens perform better; however, many tested specimens did fail by a macroscopic crack initiated within the transition zone due to stress concentration between the tab and gauge sections.

### 3.1. Material Characterization

The material was expected to be elastic. We provide the stress-strain curve of ISO 527-2 (performed with a ramp speed of 2 mm/min) in Figure 8. A linear elastic stress-strain response is observed up to 20 MPa and between 1 to 1.5% strain. A bare-eye visual inspection qualifies the cross-section of failure as a brittle crack. No plasticity occurred in the specimens. In order to determine its possible viscoelastic character, several uniaxial tests were performed on ISO 527-2 specimens with the aforementioned (displacement controlled) equipment. The maximum force was chosen by performing several measurements, and we show herein results up to 20 MPa equivalent stress. The strain rate dependencies are compiled in Table 7 with their corresponding stress-strain curves in Figure 9.

Here, all specimens were printed by the Slicing B parameters and all tests were carried out at room temperature. Evidently, no rate effects are visible at room temperature for a great range of strain rates in several orders. Therefore, we explain the stress-strain behavior after 1.5% of strain by geometric nonlinearities under large displacement.

For determining the linear elastic properties, additional uniaxial tensile tests were performed until failure. For all geometries with both slicing approaches, qualitatively, results indicate that the material shows a linear elastic response until failure at the room temperature. For a quantitative assessment, we used the slope of the stress-strain curve between 0.05 and 0.25% to determine Young’s modulus, ultimate tensile strength, and for all tests performed for different geometries are demonstrated in Figure 10.

Relatively small errors denote an adequate repeatability owing to the chosen process parameters. Below, we provide the equation for the error assessments,
(4)$$R=\frac{S}{m}\times 100=\frac{\sqrt{\frac{\sum_{i=1}^{n}\left(x_{i}-\tilde{x}\right)}{n-1}}}{\sum_{i=1}^{n}x_{i}/n}\times 100,$$
where the relative standard error, *R* in percent, is obtained by the standard deviation, *S*, and the arithmetic mean of all samples, *m*. By $x_{i}$ we denote the value of the *i*th point in the data set, where $\tilde{x}$ is the mean value of all samples and *n* is the total number of data points. Relative errors of the elasticity modulus and maximum stress are quantified in Table 8. By comparing the Slicing A and B settings, obviously the Slicing A performs better than the Slicing B regarding the obtained errors less than 2%. Figure 10 (top) and Table 8 (left) show that deviation is small in determined Young’s moduli among all specimen types. However, there are slightly higher deviations in maximum stress results with more errors in Figure 10 (bottom) and Table 8 (right). As seen from Table 8, slicing settings have a more significant impact on the accuracy than the chosen specimen type according to ASTM or ISO. Hence, we conclude that all specimen types are adequate for obtaining Young’s modulus. However, this fact changes in the case of the ultimate strength, as the error range is affected by the specimen type as well as the slicing approach. As expected, the effective mechanical properties of 3D-printed polymers are less than molded structures [68,69]. Processing conditions like low interlayer adhesion and occurring porosity within the structure are understood to be the main reasons for this outcome.

Premature failure was observed among almost all specimen types. Failures of ASTM D3039 and ASTM D3039 angles occurred in the acceptable areas (within the gauge section). ISO527-2 and ISO-modified specimens had more premature failures in non-acceptable areas, such as the shoulder (edge) sections. The Slicing A achieved shifting of the position of the failure to the gauge section, especially in ISO and ISO-modified specimens, but not in ASTM D3039 and ASTM D3039 angle specimens. Among them, the best improvement regarding the position of failure was observed in ISO 527-2 and ISO-modified specimens.

Tabs are necessary to provide an adequate clamping and curvatures are of interest to distribute the stress appropriately. However, slicing may cause some problems in the transition zone between clamping edges and gauge. As observed in many cases, the slicing software begins the production from the clamped edges and finishes these areas before continuing to the rest. This situation is depicted in Figure 3b as a result of the Slicing B, where three edges were produced initially. Obviously, the right side of the specimen had different characteristics than the left side caused by a heterogeneous solidification. The rods and layers did not connect to each other very well because of the temperature difference on the solidified layer and the deposited melt. Thus, molecular diffusion decreased at the interface between the edges and the main parts due to the low thermal energy. Experimental results proved that this approach was one of the reasons of premature failures. Usually, ISO527-2 and ISO-modified specimens were failed in those areas. Failure behavior of specimens with the Slicing B are depicted in Figure 11a,c (top).

On the contrary, in the Slicing A, the CAD model was sliced from one edge until the end of the opposite edge in one line. This production route is depicted in Figure 3a. Apparently, the specimens resulted by the Slicing A had better interlayer and interrod bonding due to homogeneous solidification than the specimens crafted by the Slicing B. The Slicing B resulted in the onset of failure within (near the end of) the transition zone as seen in Figure 11a. Fortunately, the Slicing A shifted the onset to the gauge regime as seen in Figure 11b.

### 3.2. Geometry Effect

The geometry of specimens fails to have a significant effect on mechanical properties; however, we have observed that the position of failure did depend on the chosen structure. Stress distributions of FEA simulations are provided in Figure 12.

As also mentioned in its corresponding standard [58], ASTM D3039 and ASTM D3039 angle specimens did fail near the tab sections, although this situation was undesired. According to the suggestion in ASTM D3039, more than 40% of failures near the clamping end set the design in question. The design of ASTM D3039 has been realized for fiber-reinforced composites such that the localization of stress is different than herein. We schematically show this situation in Figure 13.

ISO 527-2 specimens have been specifically designed for molding and extrusion plastics. In the case of 3D printing, specimens’ curvatures between tabs and gauge lengths provide a force distribution possibly not aligned with the used slicing approach. Hence, structure undergoes stress concentration leading to premature failure within the transition zone. Moreover, if the specimens are produced with contour lines, there may be gaps between contours and infills. This heterogeneity may lead to stress concentrations, too. Increased curvatures in ISO-modified specimens resulted more adequate and reliable positioning of failure in repeated tests. Thus, we conclude that the suggested ISO-modified geometry is to be used for characterization of mechanical response in structures manufactured by 3D printing based on fused deposition modeling.

## 4. Conclusions

In this study, the mechanical characterization of polymers has been established in fused deposition modeling, a common 3D printing technique. Four different types of geometries were investigated for uniaxial tensile experiments.

The default slicing settings were analyzed and modified in order to obtain a unidirectional structure to be used in a uniaxial tensile test. Two amendments are proposed:Excess travel lines were eliminated by using a newly developed printing path approach. The new travel path did not affect the unidirectional configurations of the specimens. In this way, premature failures have been circumvented. A more homogeneous structure has been achieved at the macroscale.An adequate slicing sequence was developed and examined. The amount of weak bond areas were reduced by this new approach. In this way, better solidification and bond formation have been accomplished, leading to a homogeneous structure at the microscale.

All these improvements have been achieved by modifying process parameters suggested by the slicer software. Different slicing approaches have been compared in order to quantify the amendment, as follows:We performed tensile tests and found that the used PETG material was linear elastic and performed a brittle fracture at room temperature (below the glass transition temperature).We emphasize that the slicing technique influences the performance of the final product significantly, such that the tensile test may lead to inaccurate results caused by a premature failure. We have discussed possible slicing techniques, especially for uniaxial tensile testing, and demonstrated the strength of an adequate approach leading to adequate repeatability, as well as a small deviation in results.We emphasize the importance of the slicing settings by showing the failure along printing directions in tensile experiments.For a better understanding of premature failure, a digital twin of laboratory tests has been realized by finite element method analysis. The same geometries have been implemented for the uniaxial test, leading to stress concentration around the onset of the failure. Based on this observation, a simple yet efficient design change is used in ISO 527-2 and ASTM D3039 specimens.The flat shape suggested in ASTM D3039 is appropriate for 3D printing; however, the transition causes a stress concentration leading to premature failures, as demonstrated in both the experimental and FEM results.ISO527-2 and ISO-modified performs better, especially by tuning the slicing approach as demonstrated herein. By using computations, we have suggested a modification for a better performance regarding repeatability and accuracy in uniaxial tensile tests.

## Figures and Tables

**Figure 1 materials-14-02556-f001:**
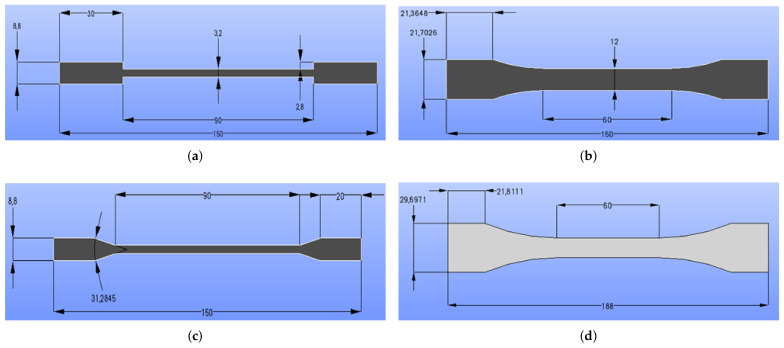
Tensile test specimen geometries and their specifications in mm: (**a**) ASTM D3039; (**b**) ISO527-2; (**c**) ASTM D3039 angle and (**d**) ISO-modified.

**Figure 2 materials-14-02556-f002:**
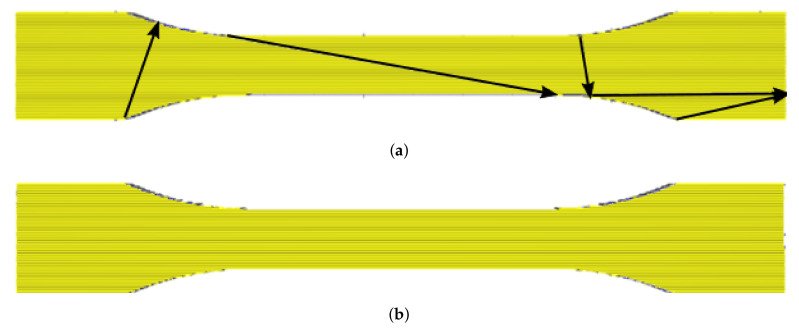
Simulation of production with two travel configuration setups by using arrows visualizing the travel of the nozzle. (**a**) Default setup with parameters from Table 4, black lines are travel lines, also depositing material, indicating weak points in the specimen. (**b**) Optimized setup, the travel lines are along the outer surface of the specimen.

**Figure 3 materials-14-02556-f003:**
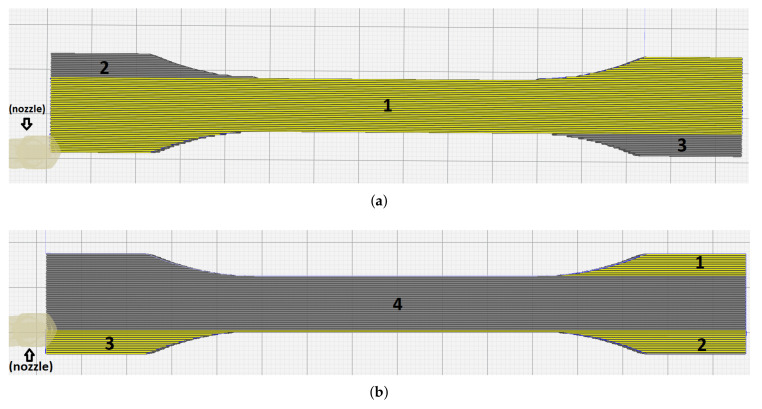
Simulation by using two different slicing approaches. (**a**) Newly optimized Slicing A and (**b**) Current Slicing B.

**Figure 4 materials-14-02556-f004:**
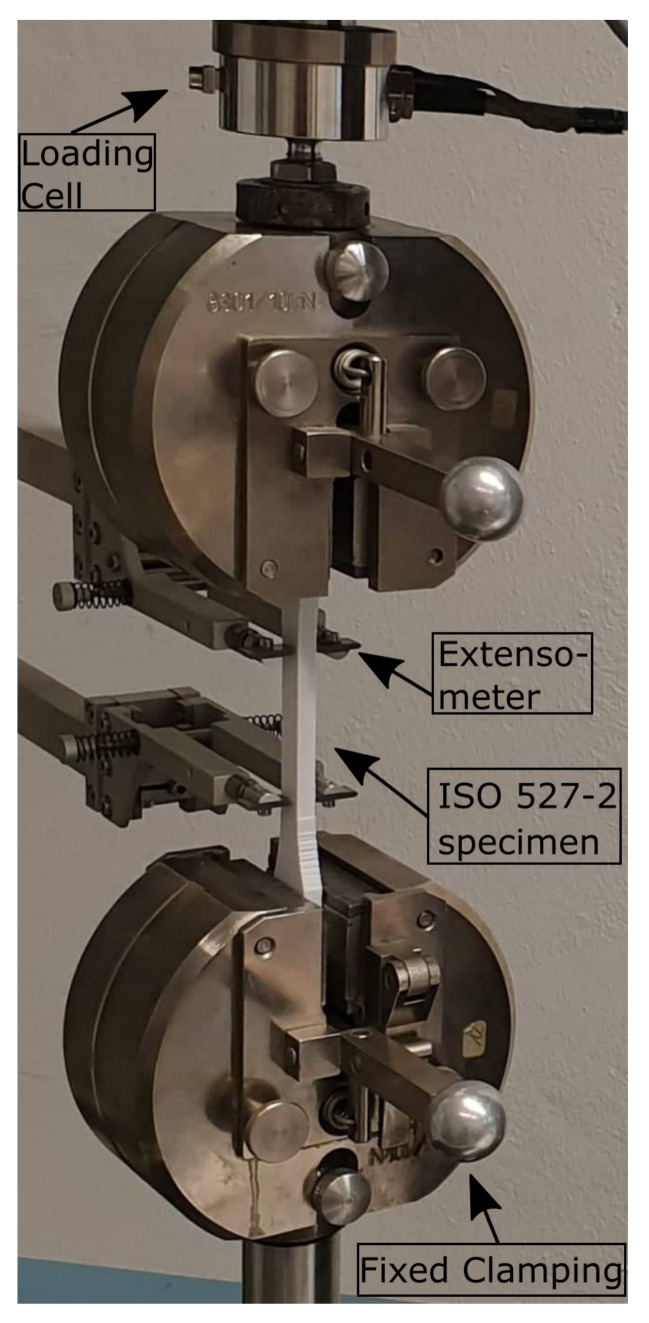
Initial state of an ISO 527-2 specimen clamped into a Zwick 1446 testing device with an extensometer (left) and a loading cell (top). One mounting side is fixed (bottom), while displacement is applied vertically through the upper mounting side.

**Figure 5 materials-14-02556-f005:**
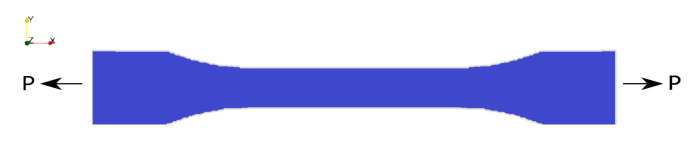
An illustration of the ISO 527-2 CAD model with boundary conditions.

**Figure 6 materials-14-02556-f006:**
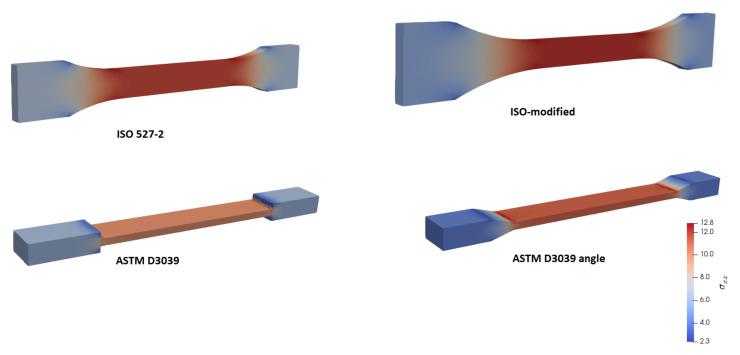
Simulation of uniaxial tensile tests, stress distribution, $\sigma_{xx}$, (in color) resulted in all investigated geometries.

**Figure 7 materials-14-02556-f007:**
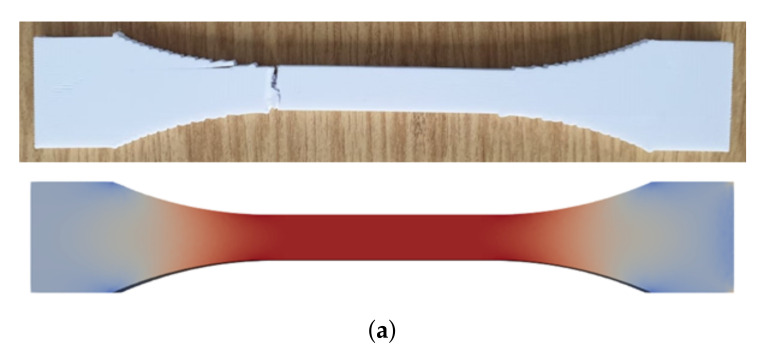
Experimental results (top) showing a brittle failure at a position directly related to the stress distribution verified by computations (bottom). (**a**) For the ISO-modified specimen. (**b**) For the ASTM D3039 specimen (zoom to the clamped end).

**Figure 8 materials-14-02556-f008:**
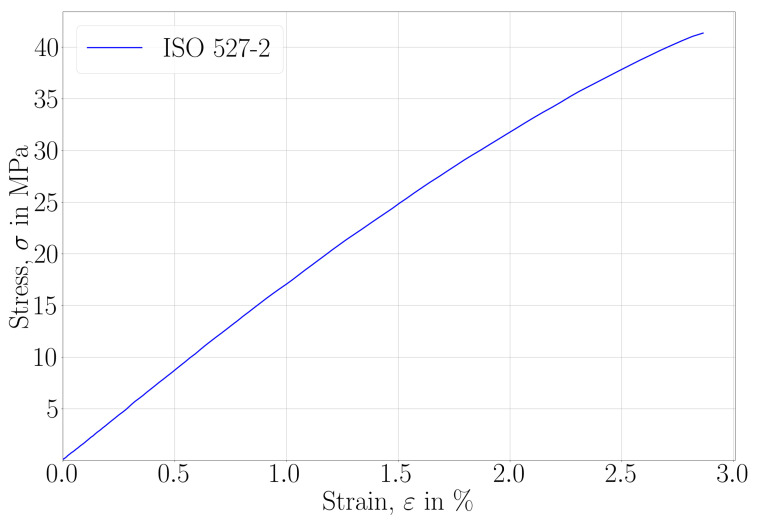
Stress-strain curves recorded in uniaxial tensile tests until failure with a ramp speed of 2 mm/min.

**Figure 9 materials-14-02556-f009:**
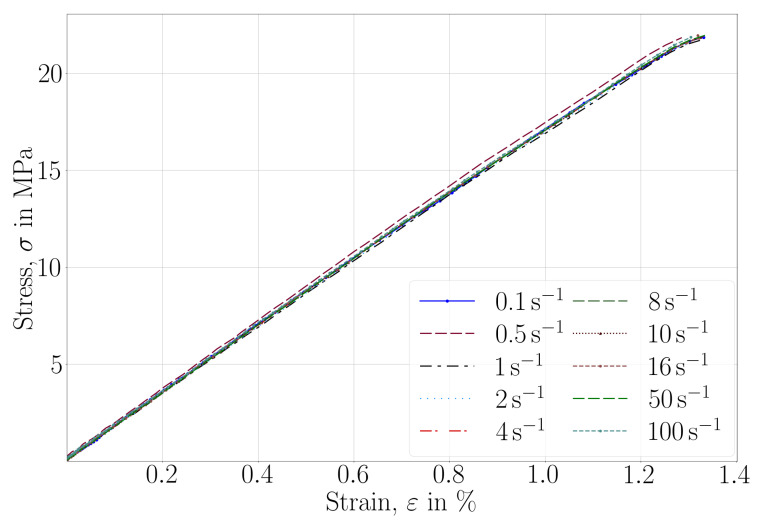
Stress-strain curves recorded in uniaxial tensile tests until 20 MPa.

**Figure 10 materials-14-02556-f010:**
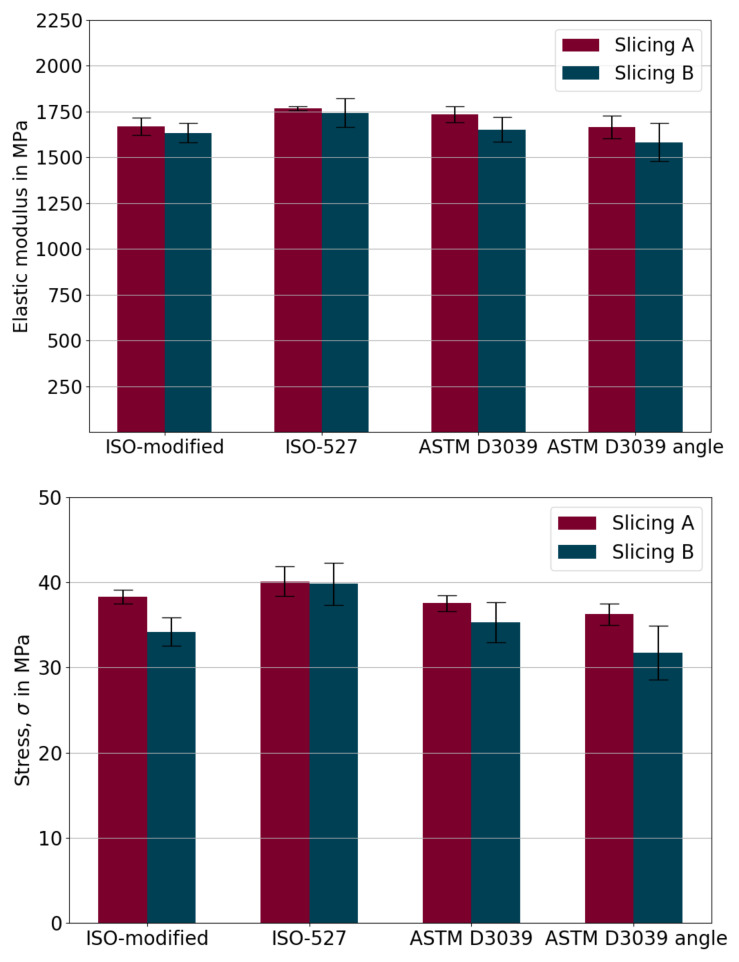
Quantitative comparison of the geometry effect in the case of tensile tests. Arithmetic mean Young’s modulus, *E*, and maximum stress before failure.

**Figure 11 materials-14-02556-f011:**
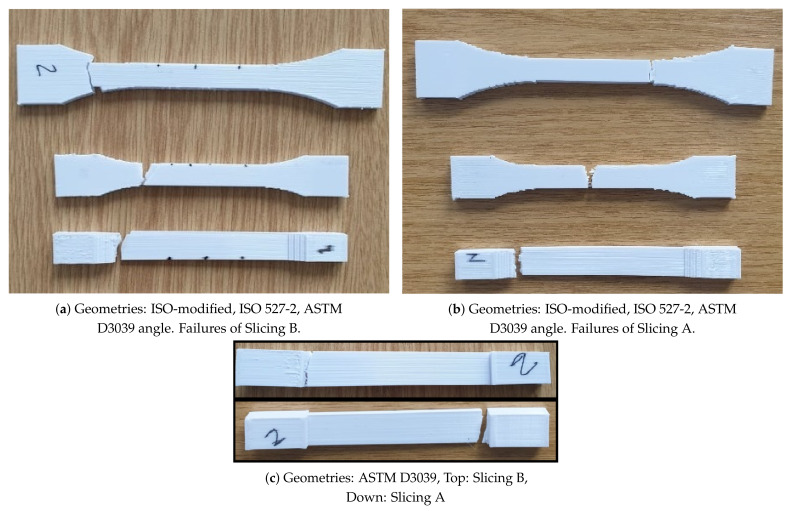
Pictures of failures according to geometry types and slicing approaches. The specimens which have been broken off out of the extensometer limit are provided only for visual observations. Their results are not included.

**Figure 12 materials-14-02556-f012:**
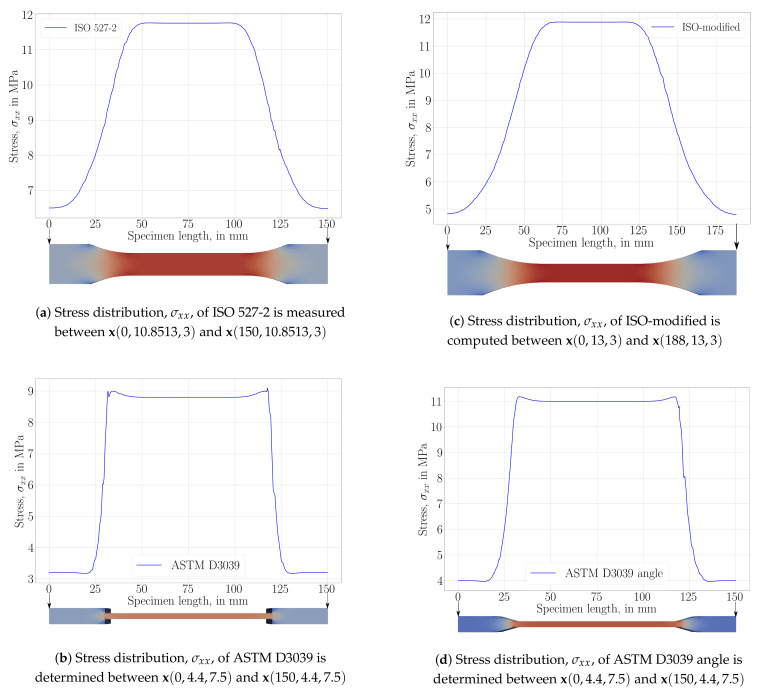
Stress results, $\sigma_{xx}$, of uniaxial tensile test simulations of different specimen geometries: (**a**) ISO527-2, (**b**) ASTM D3039, (**c**) ISO-modified and (**d**) ASTM D3039 angle.

**Figure 13 materials-14-02556-f013:**
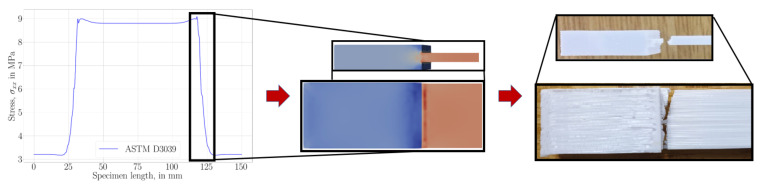
Schematic illustration of the stress concentration on ASTM D3039 specimens. Sharp stress increase is measured and visualized by finite element simulations (**left**). This outcome is validated with experimental investigations (**right**).

**Table 1 materials-14-02556-t001:** Specimen specifications.

Description	ASTM D3039	ASTM D3039 Angle	ISO527-2	ISO-Modified
Tab length in mm	30	20	21.4	21.8
Tab thickness in mm	2.8	2.8	-	-
Thickness in mm	3.2	3.2	6	6
Length in mm	150	150	150	188
Width in mm	15	15	21.7	29.7
Gauge length in mm	90	90	60	60
Angle	-	31.28°	R60	R105

**Table 2 materials-14-02556-t002:** Process parameters of 3D printing.

Parameter	Value	Unit
Layer thickness	0.3	mm
Layer width	0.4	mm
Print speed	55	mm/s
Initial layer speed	40	mm/s
Print acceleration	4000	mm/s^2^
Printing temperature	250	°C
Printing temperature initial layer	255	°C
Final printing temperature	240	°C
Bed temperature	70	°C

**Table 3 materials-14-02556-t003:** Material properties of PETG.

	Value	Unit	Method
Mass density	1.27	g/cm^3^	ASTM D792
Elongation at break	70	%	ASTM D638
Tensile strength at break	26	MPa	ASTM D638
Flexural modulus	2150	MPa	ASTM D790
Melting point	200−230	°C	ASTM D3418
Heat distortion temperature	74	°C	ASTM D648

**Table 4 materials-14-02556-t004:** Standard and modified travel configurations in Cura.

Parameter	Standard	Modified
Combining mode	All	Not in Skin
Max comb. distance with no retract	0 mm	100 mm
Avoid printed parts when traveling	✓	✓
Travel avoid distance	3 mm	10 mm
Layer Start X	213.0 mm	200.0 mm
Layer Start Y	198.0 mm	200.0 mm
Z hop when retracted	✓	✓
Z hop only over printed parts	✓	✓
Z hop height	2 mm	5 mm

**Table 5 materials-14-02556-t005:** Specimen coordinates of Slicing A.

Specimen Positions	*x*	*y*
First specimen	−11.5385	−67.8167
Second specimen	−11.5385	−2.9837
Third specimen	−11.5385	61.8167

**Table 6 materials-14-02556-t006:** Experimental design with different specimen types and slicing approaches, all produced and tested on six specimens to obtain a statistical confidence interval.

Standard Name	Slicing A	Slicing B
ISO-modified	6 specimens	6 specimens
ISO 527-2	6 specimens	6 specimens
ASTM D3039	6 specimens	6 specimens
ASTM D3039 angle	6 specimens	6 specimens

**Table 7 materials-14-02556-t007:** Strain rate-dependency, uniaxial tensile test results, normal stress at two normal strain values, $\sigma_{0.05}=\sigma \;\left(\varepsilon =0.05\%\right)$ and $\sigma_{0.25}=\sigma \;\left(\varepsilon =0.25\%\right)$, are used for determining the Young’s modulus, $E_{t}$.

	σ (0–20 MPa)		
Strain-Rate	$\sigma_{0.05}$ in MPa	$\sigma_{0.25}$ in MPa	$E_{t}$ in MPa
0.1	0.992	4.228	1796.373
0.5	1.109	4.458	1751.313
1.0	0.958	4.300	1702.541
2.0	1.041	4.280	1729.479
4.0	1.066	4.339	1738.880
8.0	0.970	4.379	1742.779
10.0	1.057	4.360	1738.423
16.0	0.949	4.331	1725.279
50.0	0.928	4.351	1735.809
100.0	1.080	4.263	1714.809

**Table 8 materials-14-02556-t008:** Relative errors of elasticity modulus and max stress.

	Relative Error of Elasticity Modulus in %		Relative Error of Max Stress in %	
Specimen Type/Slicing Approach	Slicing A	Slicing B	Slicing A	Slicing B
ISO-modified	2.851	3.305	2.179	4.945
ISO 527-2	0.604	4.462	4.331	6.293
ASTM D3039	2.480	4.087	2.566	6.644
ASTM D3039 angle	3.681	6.548	3.560	9.932

## Data Availability

Raw data were generated at the TU Berlin, Institute of Material Science and Technology. Derived data supporting the findings of this study are available from the corresponding author (Arda Özen) upon request.

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
