# Peer review of "Optimization of Manufacturing Parameters and Tensile Specimen Geometry for Fused Deposition Modeling (FDM) 3D-Printed PETG"

_materials, 2021, doi:10.3390/ma14102556_

Round 1

Reviewer 1 Report

The article is well written and has the correct structure. Here are some small remarks.

  1. In the introduction, please write a few sentences regarding the printing time with different 3D techniques.
  2. In Figures 1,2,3,7, it is better to use letters instead of words.
  3. Figure 3, please write down what the sequence means (1,2,3). Do these numbers correspond to the text on lines 151-153? Where is area 4 in Figure 3.
  4. In point 2.2, it should be stated whether the Poisson's ratio was determined in the research.
  5. Line 198, it should be added reference to the sentence "…known as the Galerkin procedure...".
  6. Figures 8, 9, 10,12,13, please remove "in" from the axis description.
  7. In the Conclusion, the results of numerical and laboratory tests should be compared (the same strain rate was used?

Reviewer 2 Report

A preliminary lecture of the article titled “Optimization of manufacturing parameters and tensile specimen geometry for fused deposition modeling (FDM) 3-D printed polymers” offers a very impressive opinion to this reviewer. A second and more profound reading confirms by far this preliminary impression.

The authors have performed a very complete study looking to establish robust criteria for measuring mechanical properties on champions obtained by additive processing of a polymer, PETG,  and nor for whatever polymer as the title suggests.

As the authors correctly claim, the processing operations are a key aspect in the ultimate properties of the material, and in this sense, they have pointed out this aspect clearly. In fact, when a polymer specimen is tested not only the polymer itself is tested but the way it has been shaped. So, please, indicate in the title the polymer used to avoid confusion.

Finally, the authors seem to have demonstrated that the geometry of the specimen is playing a crucial role. In fact, they identify premature failure that does not enable the use of these data for the academic databases. Note, as the authors remark, that some specimens break out of the extensometer marks limits, which supports this idea.

As mentioned before, the only negative concern that this reviewer has identified is related to the material used (PETG) both in terms of properties (no information about molecular weight is provided) and the absence of another type of polymer in order to contrast if the geometry dependence varied or not with the type of polymer to be fuse deposited. In the personal opinion of this referee that the latter will make more robust the investigation, which in its actual state is more a case study than a general and universal framework for whatever polymer as may be wrongly interpreted by potential readers from the title avoiding to mention that the study is focused in one polymer in particular. So, this reviewer considers it mandatory to include in the title the polymer really used.

Please, note that the corrections proposed are so easy as to consider recommending a MINOR REVISION but to be mandatorily performed.
